# Acute thermal stress increased enzyme activity and muscle energy distribution of yellowfin tuna

Hongyan Liu[1,2☺], Rui Yang[1,2☺], Zhengyi Fu[1,2,3], Gang Yu[1,2], Minghao Li[1], Shiming Dai[1,2], Zhenhua Ma[1,2,3]*, Humin Zong[4]*

1 Tropical Aquaculture Research and Development Center, South China Sea Fisheries Research Institute, Chinese Academy of Fishery Sciences, Sanya, China, 2 Key Laboratory of Efficient Utilization and Processing of Marine Fishery Resources of Hainan Province, Sanya Tropical Fisheries Research Institute, Sanya, China, 3 College of Science and Engineering, Flinders University, Adelaide, SA, Australia, 4 National Marine Environmental Center, Dalian, China

☺ These authors contributed equally to this work.
* zhenhua.ma@scsfri.ac.cn (ZM); hmzong@nmemc.org.cn (HZ)

**Data Availability Statement:** The original contributions presented in the study are included in the article.

**Funding:** This work was supported by Hainan Major Science and Technology Project

## Abstract

Heat is a powerful stressor for fish living in natural and artificial environments. Understanding the effects of heat stress on the physiological processes of fish is essential for better aquaculture and fisheries management. In this experiment, a heating rod was used to increase the temperature at 2°C/h to study the changes of energy allocation (CEA) and energy metabolity-related enzyme activities, including pepsin, trypsin, amylase, lipase, acid phosphatase, lactate dehydrogenase, alanine aminotransferase, glutamic oxalic aminotransferase and energy reserve (Ea), energy expenditure (ETS), in juvenile yellowfin tuna cells under acute temperature stress. The results showed that the Ea of juvenile yellowfin tuna muscles in response to high temperature (34°C) was significantly lower than that of the control (28°C), and it also increased ETS. At 6 h, CEA decreased slightly in the high-temperature group, but, the difference in CEA between 24 h and 0 h decreased. After heat stress for 6 h, the activities of acid phosphatase (ACP), lactate dehydrogenase (LDH), alanine aminotransferase (ALT) and glutamic oxalacetic transaminase (AST) increased, indicating that the metabolic rate was accelerated. After heat stress for 24 h, the activity of ALT decreased, indicating that with time elapsed, the activities of some protein metabolizing enzymes increased, and some decreased. In this study, digestive enzymes, trypsin and lipase increased gradually. After heat stress, Ea and Ec change significantly. Yellowfin tuna muscles use lipids in response to sharp temperature increases at high temperatures, red muscles respond to temperature changes by increasing energy in the early stages, but not nearly as much, and white muscles reduce lipids.

## 1. Introduction

Aquaculture has become one of the most viable strategies to extend depleted aquatic and endangered species and terminate the gap between supply and demand for aquatic seafood [1–3]. Aquaculture productivity is altered by several interconnected aspects, including the aquatic

(ZDKJ2021011); Central Public-interest Scientific Institution Basal Research Fund, CAFS (2020TD55); Central Public-Interest Scientific Institution Basal Research Fund South China Sea Fisheries Research Institute, CAFS (2021SD09); the Project of Sanya Yazhou Bay Science and Technology City (SKJC-2022-PTDX-015) and National Key R&D Program of China (2019YFD0900800). The funders had no role in study design, data collection and analysis, decision to publish, or preparation of the manuscript.

**Competing interests:** The authors have declared that no competing interests exist.

environment, diet, and the farmed stock [4–6]. Boosting these aspects is the base of sustainable aquaculture [7]. Yellowfin tuna is a kind of economically important fish, popular among the public, belonging to the mackerel family and tuna genus. The species is found in tropical and subtropical waters, mainly in the Pacific, Atlantic and Indian Oceans [8]. As a result, tuna has become an important source of income for many countries, including Japan, the European Union, and the United States [9]. However, overfishing is one of the biggest threats it will face, with numbers in decline and the development of yellowfin tuna in captivity [10]. Similarly, rising water temperatures due to human-caused climate change may also reduce the oxygen supply of Marine organisms such as tuna [11]. Because of their high oxygen demand, low dissolved oxygen is a limiting factor for tuna, limiting their activities in shallow water [12]. Therefore, it is necessary to study the biology of temperature stress in yellowfin tuna reaction, thereby adjusting feeding temperature and subsequent production costs.

Temperature is an important environmental factor in fish culture because its change can affect biochemical reactions and physiological functions. Temperatures beyond the optimum limits for a particular species can adversely affect health status and increase disease susceptibility [13]. The difference in water temperature will affect fish feeding, physiological activities, and changes in growth and metabolism indicators [14]. Tuna can adapt to temperature changes by adjusting its depth in the sea [15], but when it is cultured in captivity, it will be stressed by high temperatures due to limited space. Therefore, it is crucial to explore the changes in physiological and biochemical indexes, muscle energy allocation of yellowfin tuna under acute high-temperature conditions and analyze the response of fish to high temperatures.

Heat stress can lead to a change of balance in organisms, and fish reduce the effect of temperature through a temperature compensation mechanism [16, 17]. Changes in energy reserves can be used to observe the metabolic status of fish, including total proteins, glycogen and lipids [18]. Energy expenditure (respiratory electron Transfer System ETS) increases when stressed, and energy reserves decrease. Incorporating energy storage and energy expenditure into the energy equivalent cellular Energy Allocation (CEA) helps to assess energy costs under stressful conditions. CEA has proved to be a sensitive marker that can be used to examine the changes in the energy status of fish and other organisms under stress [19–22].

In addition to CEA, heat stress is also associated with the activity of key metabolic enzymes. The effect of heat stress on CEA is directly related to energy demand and metabolism [23]. Metabolism responds to thermal acclimation through pathways including glycolysis, digestion and protein synthesis, phosphate metabolism, [24]. Physiological and biochemical indicators include metabolic enzymes, digestive enzymes, immune function mainly changes in liver function. Digestion is a fundamental process of fish metabolism, and the degradation of nutrients in the digestive tract largely depends on the available enzymes. Trypsin, pepsin, lipase and amylase are the main digestive enzymes [25].

Here, we aim to integrate CEA (energy reserve and ETS activity), related metabolic and digestive markers to better understand the energy mechanism of yellowfin tuna in increasing water temperature. During short-term acute heat stress (24 h), muscle energy allocation and changes in the activity of digestive and metabolic enzymes were measured to account for energy requirements for basal metabolism. The data will provide insight into the physiological processes occurring at the cellular and biochemical levels and can provide recommendations for yellowfin tuna feeding regimens, particularly the temperature at which they are kept. According to previous research results, the appropriate temperature for juvenile yellowfin tuna is $28.0 \pm 1.0$°C [26]. By measuring metabolic enzymes in serum, liver, digestive enzymes in stomach and pyloric caeca and energy allocation in muscle of juvenile yellowfin tuna at 0 h, 6 h and 24 h after changes in environmental conditions, the effects of acute warming on energy

distribution and metabolism of juvenile yellowfin tuna were investigated, providing theoretical basis for further research on the body stress response caused by environmental changes. And provide reference for the practice of yellowfin tuna culture.

## 2. Materials and methods

### 2.1 Experiment design and sample collection

Juvenile yellowfin tuna was raised in an offshore sea cage near Xincun Port, Xincun Town, Lingshui County, Hainan Province, China (E108˚80'10", N18˚ 50'15'). It was then temporarily cultured for 7 days at the Tropical Aquaculture Research and Development Center of the South China Sea Fisheries Research Institute, Chinese Academy of Fishery Sciences (Lingshui, China). During the adaptation process, fresh fish (Trachurus japonicus, 4 cm × 2 cm pieces) were fed 5–8% of their body weight daily. Before the experiment, 60 fish were randomly collected and transferred to two indoor cement tanks (5 m in diameter and 2.5 m in depth) containing a water circulation system. The water temperature was 28.0 ± 0.5˚C, dissolved oxygen > 5.27 mg/L, ammonia nitrogen < 0.1 mg/L, pH 7.57 ± 0.12, salinity 32‰.

At the beginning of the experiment, 60 fish of similar size (body length 28.03 ± 1.78 cm, wet weight 503.23 ± 36.78 g) were divided into two groups, the control group and the high temperature group. The temperature was set at 28˚C (control group) and 34˚C (HT group) in a 3000 L tank. The rising speed of water temperature is 2˚C/h, and the timing started when the temperature reached 34˚C. The temperature of HT group was improved and maintained by heating rod (China Zhoushan Sensen Group Co., LTD, Zhejiang, China).

The sampling time was 0 h, 6 h and 24 h after the experiment. The exact time for recording data is 6:00 PM. When the temperature stress time reached, samples of three fish were randomly collected from each feeding tank immediately, and the selected young fish were anesthetized with 0.03% MS-222. The intestines, stomach, pyloric caeca, liver, red and white muscles and blood are removed. Blood was taken from the caudal vein. The white muscle was taken at 2 cm below the dorsal fin, and the red muscle was taken near the spine perpendicular to the dorsal fin. The sampled tissues were stored in 2 ml RNA-free test tubes in an ultra-low temperature refrigerator at -80˚C until enzyme activity was determined. The extracted blood was stored in a 2 mL centrifuge tube for 1 h and centrifuged using a high-speed refrigerated centrifuge (Gloucester, Prima PB100, England) at 4˚C and 3000 R· min$^{-1}$ for 10 min and stored at −20˚C until analysis.

### 2.2 Enzyme activity measurement

All the tissue samples were partially thawed and homogenized mechanically using a tissue homogenizer on ice. The suspensions were centrifuged according to the requirements of the kits (Nanjing Jiancheng Bioengineering Institute, Nanjing, China), and the protein content in the supernatant was determined by the BCA Protein Assay kit (A045-4-2). The activity of amylase (C016-1-1) and lipase (A054-1-1) in the intestinal tract was determined. The activity of acid phosphatase (A060-2-1) and alanine aminotransferase concentration (C009-2-1) in serum were determined. The activities of acid phosphatase and alanine aminotransferase concentration in liver were determined. Pepsin (A080-1-1) in the stomach and trypsin (A080-2-2) in the pyloric caeca were measured.

### 2.3 Determination of energy distribution

CEA was used to examine stress-induced metabolic changes in juvenile yellowfin tuna and calculate the ratio of available energy (Ea, including total protein, lipid, and carbohydrate

content) to energy expenditure (Ec, determined based on ETS activity measurements). The collected muscle tissue was homogenized according to the procedure on the kit. The homogenate was used to quantify energy reserves, including protein, lipid, carbohydrate, and ETS activity.

The red and white muscles were given muscle energy allocation, including protein, fat (10110.2), glycogen (A043-1-1) and ETS. The lipid quantification kit (10110.2 v.A) was used by Shanghai Haring Biotechnology Co., LTD., and the ETS was determined by the methods in other literature [27]. Except for fat and ETS, all tests were performed using a commercial kit (Nanjing Jiancheng Bioengineering Institute, Nanjing, China) in triplicates. At the end of the assay, the corresponding enthalpies of combustion were used to convert the three energy reserves to energy equivalents of 24,000 mj·mg$^{-1}$ protein, 39,500 mj·mg$^{-1}$ lipid and 17,500 mj·mg$^{-1}$ carbohydrate [28].

ETS activity was determined according to the method described by Bin Wen and Shi-Rong Jin [27]. First, red and white muscles were homogenized in a homogenized buffer consisting of 0.1 M Tris-HCl, 0.2% (w·v$^{-1}$) Triton X-100, pH 8.5, 75 μM MgSO$_4$, and 0.15% (w·v$^{-1}$) polyvinylpyrrolidone. The homogenized solution was centrifuged (4˚C, 3000g, 10min), 100 μL supernatant was added to 300 μL substrate solution, which consisted of 0.1M Tris-HCl, 0.2% Triton X100, pH 8.5, 1.7 mM NADH, 0.25 mM NADPH. 100 μL8 mM iodonitrotetrazole was added for 10 min at 20˚C, and 100 μL quenching solution (50% formalin and 1M H$_3$PO$_4$) was added to stop the reaction. The resulting absorbance is measured at 490 nm. Use e = 15,900·M· cm$^{-1}$ to calculate the dirty amount formed. The formation of 2 mmol of methylate is equal to the consumption of 1 mmol O$_2$ in ETS. For an average mixture of proteins, lipids, and carbohydrates, an oxygen enthalpy equivalent of 484 kJ·mol$^{-1}$ O$_2$ was used to convert the resulting energy expenditure into caloric value. CEA (Ea·Ec$^{-1}$) at each time point (0 h, 6 h, and 24 h) was determined, and the total Ea in muscle tissue at each temperature (28˚C and 34˚C) was given by the sum of the three energy reserve caloric values. Ec was calculated as converting ETS activity into caloric value. Energy allocation = Ea·Ec$^{-1}$, Ea = protein + lipid + carbohydrate (mJ·mg$^{-1}$ sample) Ec = ETS activity (mJ/mg sample).

## 2.4 Calculations and statistical analysis

Excel 2021 software was used for data sorting, Origin 2021 for drawing, and SPSS26.0 software for significant difference analysis. Comparisons between different groups were conducted by independent T-test, and significant difference was set at $P < 0.05$.

## 2.5 Ethics statement

The experiment complied with the regulations and guidelines established by the Animal Care and Use Committee of South China Sea fisheries Research Institute, Chinese Academy of Fishery Sciences.

## 3. Results

### 3.1 Changes in digestive enzyme indexes under an acute temperature rise

The activity of digestive enzymes and related indexes of yellowfin tuna underwent a series of changes after acute heating stress. Under the condition of acute heating, the pepsin activity in the stomach of juvenile yellowfin tuna (Fig 1A) first decreased and then increased with time, and there was no significant difference at each time point ($P > 0.05$). At 0 h and 24 h, pepsin in the high-temperature group was higher than in the control group but lower than in the control group at 6 h (Fig 1A). Under the condition of acute heating, the pyloric caeca trypsin

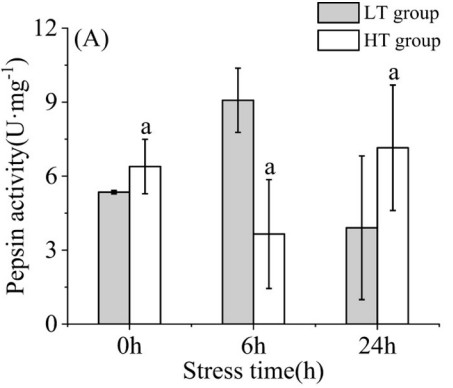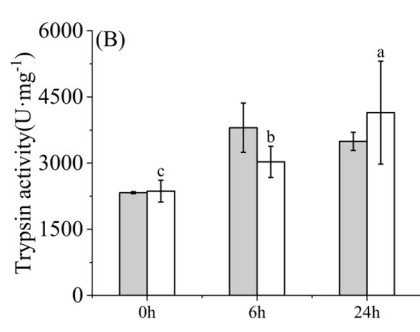

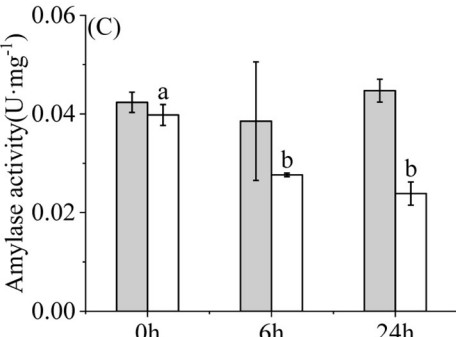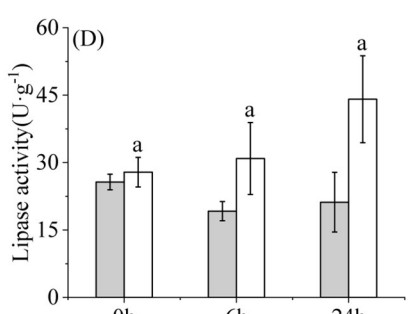

**Fig 1. Changes of pepsin activity of stomach, trypsin activity of pyloric caeca and amylase, lipase in of the intestinal tract of young yellowfin tuna under acute warming.** Different letters and "▲" indicate significant differences ($P < 0.05$). "LT group" represents the control group and "HT group" represents the high-temperature group.

activity of juvenile yellowfin tuna (Fig 1B) showed a gradually increasing trend with time, and there were significant differences at each time point ($P < 0.05$). There was no significant difference between the experimental group and the control group at 0 h, 6 h and 24 h ($P > 0.05$). At 0 h and 24 h, the trypsin activity concentration in the high-temperature group was higher than that at 28˚C, but lower than that at 6 h (Fig 1B). Intestinal amylase activity in the high-temperature group (Fig 1C) gradually decreased. There was no significant difference in amylase activity between 6 h and 24 h ($P > 0.05$), but there was a significant difference between 0 h and 6 h and 24 h ($P < 0.05$). At 0 h, 6 h and 24 h, amylase activity in the high-temperature group was lower than 28˚C (Fig 1C). At 0, 6 and 24 h, there was no significant difference in intestinal amylase between high temperature group and 28˚C ($P > 0.05$). The activity of intestinal lipase in high temperature group (Fig 1D) showed a trend of increasing gradually with the extension of time, and there was no significant difference at each time point ($P > 0.05$). At 0, 6 and 24 h, the intestinal amylase activity in the high-temperature group was higher than that at 28˚C (Fig 1D), but there was no significant difference between the high-temperature group and 28˚C ($P > 0.05$).

## 3.2 Effects of acute high temperature on metabolic enzyme activities in liver tissue

Under the condition of acute warming, the activity of liver acid phosphatase in high temperature group (Fig 2A) increased firstly and then decreased with time, and there was no

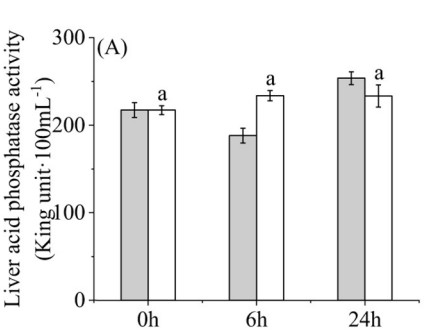

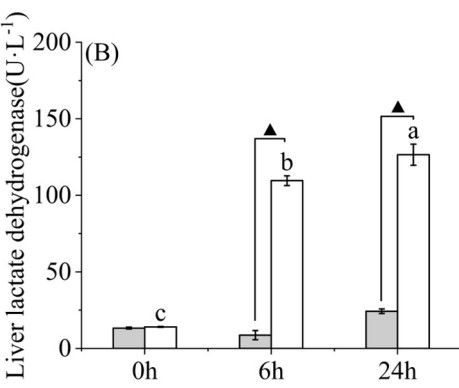

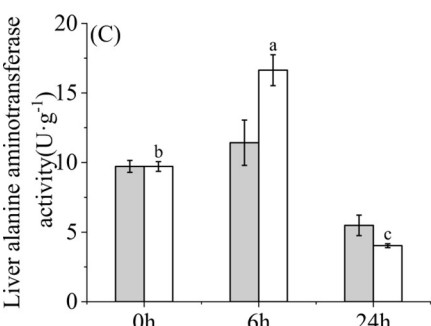

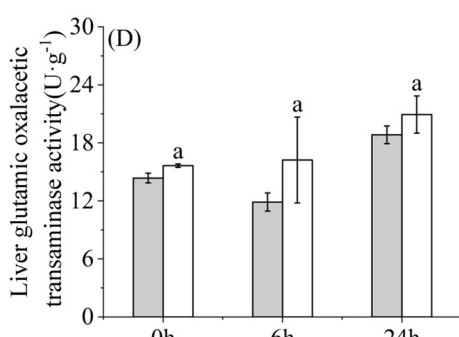

**Fig 2. Metabolic enzyme activities in liver tissue of yellowfin tuna after 24 h acute high temperature.** (A) acid phosphatase (ACP), (B) lactate dehydrogenase (LDH), (C) alanine aminotransferase (ALT), and (D) aspartate aminotransferase (AST). Different lowercase letters indicate significant temperature difference at each time point ($P < 0.05$). Those marked with "▲" represented significant difference between the control group and the high-temperature group at the same time point ($P < 0.05$). "LT group" represents the control group, and "HT group" represents the high-temperature group.

significant difference among all time points ($P > 0.05$). At 0 h and 6 h, liver acid phosphatase in the high-temperature group was higher than 28°C, and lower than 28°C at 24 h (Fig 2A). At 0, 6 and 24 h, there was no significant difference in liver acid phosphatase activity between high temperature group and 28°C ($P > 0.05$). The liver lactate dehydrogenase activity (Fig 2B) in the high-temperature group was gradually increased with time, and there were significant differences among 0, 6 and 24 h ($P < 0.05$). At 0, 6 and 24 h, the liver lactate dehydrogenase activity in the high-temperature group was higher than that at 28°C (Fig 2B). At 6 h and 24 h, there were significant differences in liver lactate dehydrogenase between high-temperature group and 28°C ($P < 0.05$). The liver alanine aminotransferase activity in the high-temperature group (Fig 2C) first increased and then decreased with time, and there were significant differences between 0, 6 and 24 h ($P < 0.05$). At 0 h and 6 h, the activity of liver alanine aminotransferase in the high-temperature group was higher than 28°C and lower than 28°C at 24 h (Fig 2C). At 0, 6 and 24 h, there was no significant difference in liver alanine aminotransferase between high-temperature group and 28°C ($P > 0.05$). Liver glutamic oxalacetic transaminase activity (Fig 2D) in high temperature group showed a gradually increasing trend with time, and there was no significant difference among 0, 6 and 24 h ($P > 0.05$). At 0, 6 and 24 h, the activity of liver glutamic oxalacetic transaminase activity in high-temperature group was higher than that at 28°C, and there was no significant difference between high-temperature group and 28°C at 0, 6 and 24 h ($P > 0.05$) (Fig 2D).

## 3.3 Effect of high temperature on muscle energy allocation

Acute hyperthermia significantly affected the distribution of proteins, lipids, and carbohydrates in both red and white muscles. The protein content of red muscle and white muscle of yellowfin tuna exposed to temperature increased with time and was significantly lower at 0 h than at 6 h, and there was no significant difference between 6 h and 24 h ($P > 0.05$) (Fig 3A and 3G). Compared with the control group, the protein content in fish muscle in the high-temperature group at 0 h and 6 h was lower than that in the control group, but higher at 24 h, and there was a significant difference between the two groups at 6 h and 24 h (Fig 3A and 3G).

Lipid content in red muscle and white muscle of yellowfin tuna exposed to temperature increase first decreased and then increased with time, with significant differences between 0, 6 and 24 h ($P < 0.05$) (Fig 3B and 3H). Compared with the control group, lipid content in the red muscle of fish body in the high-temperature group at 0 h and 6 h was lower than that in the control group at 0 h, but higher at 6 h and 24 h; lipid content in the white muscle was higher at 0 h and lower than that in the control group at 6 h and 24 h (Fig 3B and 3H). At 6 h and 24 h, the lipid content in red muscle and white muscle of the high-temperature group was significantly different from that of the control group ($P < 0.05$).

The carbohydrate content in the red muscle of yellowfin tuna exposed to temperature increased first and then decreased over time, while the carbohydrate content in the white muscle increased gradually. The carbohydrate content in red muscle was significantly different at 0 h and 6 h ($P < 0.05$), but there was no significant difference between 0, 6 and 24 h. There was no significant difference between 6 h and 24 h ($P > 0.05$), and there was a significant difference between 0 h and 6 h ($P < 0.05$) (Fig 2C and 2I). Compared with the control group,

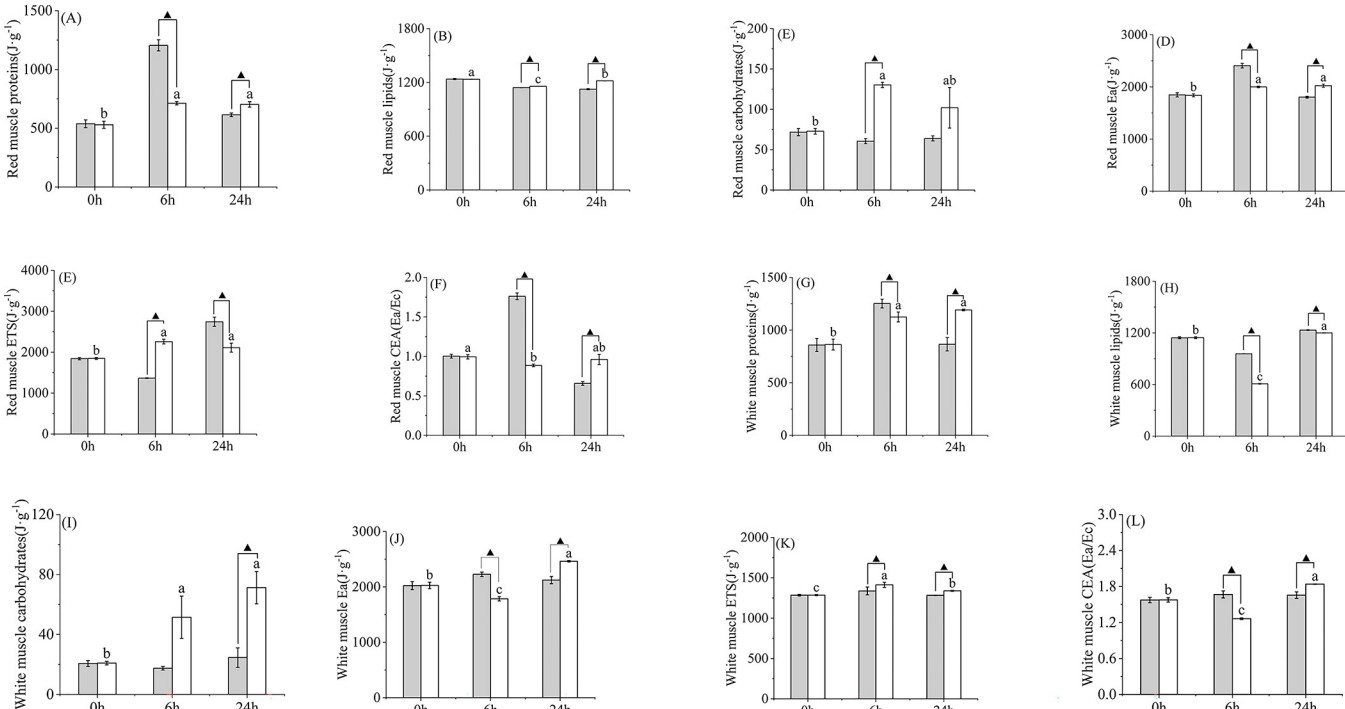

**Fig 3. Energy reserves in muscle tissue of yellowfin tuna exposed to both temperatures for 24 h.** (A-F) red muscle, (G-L) white muscle. Different lowercase letters indicate significant differences at each time point ($P < 0.05$). Those marked with "▲" represented significant differences between the control group and the high-temperature group at the same time point ($P < 0.05$). "LT group" represents the control group, and "HT group" represents the high-temperature group.

carbohydrate content in red muscle and white muscle in the high-temperature group at 0, 6 and 24 h was higher than that in the control group; at 6 h, carbohydrate content in red muscle was significantly different between the high-temperature group and the control group, and at 24 h white muscle was significantly different between the two groups ($P < 0.05$) (Fig 3C and 3I).

Temperature and time had significant effects on total Ea (Fig 3D and 3J). Ea in the red muscle of yellowfin tuna exposed to temperature increased gradually over time, while Ea in the white muscle decreased first and then increased. Ea content in red muscle was significantly different at 0 h and 6 h ($P < 0.05$), but was not significantly different between 6 h and 24 h ($P > 0.05$). There were significant differences in white muscle at 0, 6 and 24 h ($P < 0.05$) (Fig 3D and 3J). Compared with the control group, the content of Ea in the red and white muscles of fish body in the high temperature group was lower than that in the control group at 0 h and 6 h, and higher than that in the control group at 24 h. Ea in red muscle and white muscle at 6 h and 24 h was significantly different between the high-temperature group and the control group ($P < 0.05$) (Fig 3D and 3J).

The ETS content in the red and white muscles of yellowfin tuna exposed to temperature increased first and then decreased over time. ETS content in red muscle was significantly different at 0 h and 6 h ($P < 0.05$), but was not significantly different between 24 h and 6 h. The intramuscular content of white was significantly different at 0, 6 and 24 h ($P < 0.05$) (Fig 3E and 3K). Compared with the control group, ETS content in the red muscle at 0 h and 6 h in the high-temperature group was higher than that in the control group but lower than that in the control group at 24 h. The content of ETS in white muscle in 0, 6 and 24 h high temperature groups was higher than that in the control group. At 6 h and 24 h, ETS content in red and white muscle was significantly different between the high-temperature group and the control group (Fig 3E and 3K).

CEA in red muscle and white muscle of yellowfin tuna exposed to temperature increase first decreased and then increased over time. CEA of red muscle had a significant difference between 0 h and 6 h ($P < 0.05$), but had no significant difference between 0, 6 and 24 h. There were significant differences in white muscle at 0, 6 and 24 h ($P < 0.05$) (Fig 3F and 3L). Compared with the control group, CEA in the red muscle at 0 h and 6 h in the high-temperature group was lower than that in the control group, but higher than that in the control group at 24 h. The CEA in white muscle in 0 h and 24 h high temperature groups was higher than in the control group but lower than in the control group at 6 h. At 6 h and 24 h, CEA in red and white muscle showed significant differences between the high-temperature group and the control group (Fig 3F and 3L).

## 4. Discussion

### 4.1 Changes of muscle energy allocation indexes under an acute temperature rise

In this study, to investigate changes in energy reserve and metabolic response of yellowfin tuna under high temperature stress, CEA and related enzymes were examined at 0 h, 6 h and 24 h adaptation to high temperature. Measurements of these indicators can provide insights into the energy state and physiological processes of yellowfin tuna in response to heat stress.

It is well known that in addition to the energy stored in the body of animals for metabolism of daily activities (such as swimming and feeding), the remaining energy is usually stored in the liver or muscle tissue in the form of lipids [29]. Lipids and carbohydrates are the most directly available sources of energy [20, 30], while proteins are the last energy reserves to be used when exposed to stress [31]. In this study, the protein content in the muscle of the high-

temperature group first increased and then gradually stabilized, the fat content in the muscle significantly decreased and then increased, and the carbohydrate content significantly increased. There was no reduction in carbohydrates, but only in fat, suggesting that fat may be consumed more directly than carbohydrates and protein. ETS activity (Ec) represents the amount of oxygen consumption that occurs when all enzymes are functioning at their maximum [27] and increases significantly with decreasing temperature. Both the initial energy fraction Ea (mainly lipids and carbohydrates) and Ec in muscle tissue changed significantly at elevated temperatures. Results At 6 h, CEA decreased significantly with the increase of temperature, which was the result of the significant decrease of available energy (lipids and carbohydrates) and the significant increase of energy consumption. Although CEA in red muscle also showed a significant decrease trend at 6 h, Ea and ETS increased significantly, but Ea increased less than ETS. These results confirm that yellowfin tuna muscles use lipids in response to acute temperature increases at high temperatures, red muscle responds to temperature changes by increasing energy in the early stages, but not nearly as much as it expended, and white muscle decreases lipids. After 24 h of adaptation, muscle lipids recovered from the initial consumption and CEA also increased, indicating that yellowfin tuna can undergo necessary physiological adjustment involved in acute heat stress within 24 h [32]. Similar results have been observed in red seabream (*Pagrus major*) [33] and sea cucumber [19] in response to heat stress.

## 4.2 Changes of energy-related enzyme activity in liver under an acute temperature rise

CEA is associated with direct energy metabolism, so the activities of key metabolic enzymes, including acid phosphatase (ACP), lactate dehydrogenase (LDH), alanine aminotransferase (ALT) and aspartate aminotransferase (AST), were examined. The results showed that there were significant differences in the activity of different enzymes in liver. The enzymes in fish liver respond to heat stress differently and have different functions, and there are differences among different enzymes. ALT and AST are related to amino acid metabolism [34]. In fish, liver is the main site of amino acid deamination [35]. In the present study, the activity of ALT and AST increased at higher temperatures, indicating the mobilization of free amino acids for energy production [36]. The increased levels of ALT and AST may be a marker of liver dysfunction, reflecting liver cell damage caused by thermal stress [37]. Alterations in enzyme activity related to cellular energy metabolism caused by acute heat stress have been reported in various fish species. For example, ALT and AST activities were increased in Red Cusk-Eel (Geypterus chilensis) liver adapted to acute heat stress [38]. The AST and ALT activities in plasma of Takifugu obscurus [39] and Scophthalmus maximus [40] were increased. In this study, yellowfin tuna exposed to elevated temperature showed increased activity of ALT and AST in liver tissue during 6 h, indicating a rapid response to acute heat stress. At 24 h, only AST increased and ALT decreased. This indicates that metabolism is accelerated at 6 h to produce energy in response to environmental changes, while at 24 h, it may slightly adapt to temperature changes, with the activity of some protein-metabolizing enzymes decreasing and some increasing.

ACP plays an important role in disease resistance. Phagocytosis of active macrophages is involved in the dissolution of dead cells [41]. When fish are subjected to short-term heat stress, it is recognized as acute stress and can enhance immunity to cope with stress by transferring energy and increasing proteins such as ACP and complement [42]. In this study, ACP activity was increased by heat treatment, indicating that hydrolysis of high-energy phosphate bonds releases phosphate ions to combat stressful conditions or high metabolic rates [43]. Similarly, ACP activity of young *Lates calcarifer* under acute temperature stress increased after exposure

to low temperature [44]. Some studies have found that the ACP activity of striped catfish (*Silurus asotus*) has no significant difference under different temperatures [45], which is the same as the results of this study, indicating that the ACP activity in the liver of yellowfin tuna is less affected by temperature.

The changes in aerobic capacity of organisms exposed to stress conditions reflect the changes in specific metabolic pathways, which gradually lead to the transformation of energy production mode. When aerobic metabolism itself cannot meet energy demand, anaerobic energy production pathways must be activated [46]. LDH and other glycolytic enzymes exist in the cytoplasm to catalyze the last step of anaerobic glycolysis, and the activity and function of this enzyme reflect the ability of anaerobic energy production to eliminate lactic acid during aerobic recovery under conditions of hypoxia, intense exercise and heat stress [47]. In this study, LDH activity showed an increasing trend with the increase of temperature and time, indicating that the balance of carbohydrates was destroyed. After the temperature rises, more lactic acid is secreted, so the lactate dehydrogenase is higher. Similarly, studies have found that turtles LDH activity is significantly increased in response to acute heat, *Pelodiscus sinensis* [46, 48]. However, fish reactivity not only depends on the timing of adaptation to these environmental variables and stresses, but may also be species-specific [46]. In related studies, it was found that LDH activity in white muscle and liver of Tilapia was significantly increased with increasing temperature when adapting to high temperature, while LDH activity in liver was on the contrary [49], which was different from the results of this study. In future ocean scenarios, a shift from aerobic to anaerobic metabolism is needed to maintain the higher energy requirements of the species.

### 4.3 Changes in digestive enzyme indexes under an acute temperature rise

After short-term exposure to two fixed temperatures, digestive enzyme activity is regulated in different ways. In related studies, it was found that the lipase activity of *Chlamys Farreri* increased significantly at higher temperature and was higher than that of other groups [50], which was consistent with the results of this experiment, suggesting that high temperatures lead to accelerated fat consumption. Pepsin is a relatively stable enzyme under heat stress [51]. In this study, there was no significant difference in the time of pepsin activity of yellowfin tuna in the high-temperature group, this is consistent with the properties of pepsin. In *Gasterosteus aculeatus*, amylase in fish with high temperature gradually decreased and was higher than that at low temperature, while trypsin did not change [52]. The changes of amylase in this study were similar to those in the *Gasterosteus aculeatus*. activity of amylase is much lower than that of other enzymes because yellowfin tuna is a carnivorous fish and the activity of amylase is extremely low. It has been reported that the low energy activation of amylase activity in different fish is an adaptation of the digestive system to temperature maintenance by producing new highly efficient isoenzymes. Trypsin increased significantly, this is different from the above studies, which may be due to the unbalanced energy distribution of three-acanthus, which damages the synthesis of trypsin [51]. This suggests that high temperatures accelerate the metabolism of organisms, in which case the energy balance shifts to maintenance rather than growth [53].

## 5. Conclusion

In conclusion, CEA indicated that juvenile yellowfin tuna liver was significantly affected by acute temperature stress, but could recover energy balance after 24 h adaptation. The increased activity of energy metabolism-related enzymes and most digestive enzymes reflects the accelerated metabolism of fish at high temperature. The results showed that yellowfin tuna had

remarkable resilience in the acute heating stress environment. In this study, the energy balance of juvenile yellowfin tuna was sensitive to temperature rise, and the influence of short-term high temperature stress was small. Therefore, in production and large-scale factory farming, drastic changes in temperature should be avoided as far as possible to reduce the frequency and practice of acute temperature stress, so as to make it a good breeding and growing environment.

## Author Contributions

**Conceptualization:** Rui Yang, Zhengyi Fu, Gang Yu, Zhenhua Ma.

**Data curation:** Rui Yang, Minghao Li.

**Formal analysis:** Rui Yang, Gang Yu.

**Funding acquisition:** Zhenhua Ma, Humin Zong.

**Investigation:** Shiming Dai.

**Methodology:** Hongyan Liu.

**Project administration:** Zhenhua Ma, Humin Zong.

**Resources:** Gang Yu, Zhenhua Ma.

**Software:** Minghao Li.

**Supervision:** Zhengyi Fu, Shiming Dai, Zhenhua Ma.

**Validation:** Zhengyi Fu, Shiming Dai.

**Visualization:** Minghao Li.

**Writing – original draft:** Hongyan Liu.

**Writing – review & editing:** Zhengyi Fu, Zhenhua Ma.

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
