## [Decision Letter · Decision Letter 0]

13 Jun 2023

PONE-D-23-13386Acute thermal stress increased enzyme activity and muscle energy distribution of yellowfin tunaPLOS ONE

Dear Dr. Ma,

Thank you for submitting your manuscript to PLOS ONE. After careful consideration, we feel that it has merit but does not fully meet PLOS ONE’s publication criteria as it currently stands. Therefore, we invite you to submit a revised version of the manuscript that addresses the points raised during the review process.

We look forward to receiving your revised manuscript.

Kind regards,

Mohammed Fouad El Basuini, Professor

Academic Editor

PLOS ONE

Journal Requirements:

  "FUNDING

This work was supported by Hainan Major Science and Technology Project 

(ZDKJ2021011); Central Public-interest Scientific Institution Basal Research Fund, 

CAFS (2020TD55); Central Public-Interest Scientific Institution Basal Research Fund 

South China Sea Fisheries Research Institute, CAFS (2021SD09); the Project of 

Sanya Yazhou Bay Science and Technology City (SKJC-2022-PTDX-015)."

   "This work was supported by Hainan Major Science and Technology Project (ZDKJ2021011); Central Public-interest Scientiﬁc Institution Basal Research Fund, CAFS (2020TD55); Central Public-Interest Scientiﬁc Institution Basal Research Fund South China Sea Fisheries Research Institute, CAFS (2021SD09); the Project of Sanya Yazhou Bay Science and Technology City (SKJC-2022-PTDX-015)."

Additional Editor Comments:

I recommend using the following paragraph at the beginning of this section:

Aquaculture has become one of the most viable strategies to extend depleted aquatic and endangered species and terminate the gap between supply and demand for aquatic seafood [1–3]. Aquaculture productivity is altered by several interconnected aspects, including the aquatic environment, diet, and the farmed stock [4–6]. Boosting these aspects is the base of sustainable aquaculture [7].

1. https://doi.org/10.1016/j.aaf.2022.03.014

2. https://doi.org/10.3390/ani12172219

3. https://doi.org/10.1016/j.aqrep.2022.101460

4. https://pubmed.ncbi.nlm.nih.gov/32507794/

5. https://pubmed.ncbi.nlm.nih.gov/32001349/

6. https://pubmed.ncbi.nlm.nih.gov/32173664/

7. https://doi.org/10.1016/j.aqrep.2022.101373

Reviewers' comments:

Reviewer's Responses to Questions

**Comments to the Author**

1. Is the manuscript technically sound, and do the data support the conclusions?

Reviewer #1: Yes

Reviewer #2: Yes

Reviewer #3: Yes

2. Has the statistical analysis been performed appropriately and rigorously? 

Reviewer #1: Yes

Reviewer #2: Yes

Reviewer #3: Yes

3. Have the authors made all data underlying the findings in their manuscript fully available?

Reviewer #1: Yes

Reviewer #2: Yes

Reviewer #3: Yes

4. Is the manuscript presented in an intelligible fashion and written in standard English?

Reviewer #1: Yes

Reviewer #2: Yes

Reviewer #3: Yes

5. Review Comments to the Author

Reviewer #1: The experiment was well designed and the paper is very information. The significant and application of the study quite lucid. The following comments should however be noted for minor revision:

1. A sentence or two describing methodology should be inserted in the Abstract

2. Line 107- the intestines, pancreas, liver and blood were removed (not are removed from the fish).

3. Bulk of the discussion is a recapitulation of the results and methodology. Discussion should be majorly interpretation of results and inferences!

4. Why is cell division phase (prophase ) brought in here?

5. Lates calcarifer being a species should be italicised, also Pelodiscus sinensis and Pelodiscus sinensi.

6. Chlamys Farreri should be written in italics-Chlamys ferreri, Gasterosteus aculeatus

7. In this study, intestinal amylase activity in the high temperature group decreased significantly over time and was lower than that in the control group, which was the same as the result of this study, trypsin increased significantly- recast this statement.

Reviewer #2: This article entitled “Acute thermal stress increased enzyme activity and muscle energy distribution of yellowfin tuna” by Liu et al. is focused on impact of heat stress on enzyme activity of yellowfin tuna. Heat is a powerful stressor for fish living in natural and artificial environments. Understanding the effects of heat stress on the physiological processes of fish is essential for better aquaculture and fisheries management. The 24 h experiment was conducted to study the changes in cellular energy allocation (CEA) and energy metabolism-related enzyme activities of juvenile yellowfin tuna (Thunnus albacares) under acute temperature stress, which significantly reduced energy reserve (Ea) in muscle from 2 °C/h to high temperature (34 °C) from control (28 °C). It also increases energy expenditure (ETS). At 6 h, CEA decreased slightly in the high-temperature group, but, the difference in CEA between 24 h and 0 h decreased. After heat stress for 6 h, the activities of acid phosphatase (ACP), lactate dehydrogenase (LDH), alanine aminotransferase (ALT) and glutamic oxalacetic transaminase (AST) increased, indicating that the metabolic rate was accelerated. After heat stress for 24 h, the activity of ALT decreased, indicating that with time elapsed, the activities of some protein metabolizing enzymes increased, and some decreased. In this study, digestive enzymes, trypsin and lipase increased gradually. After heat stress, Ea and Ec change significantly. Yellowfin tuna muscles use lipids in response to sharp temperature increases at high temperatures, red muscles respond to temperature changes by increasing energy in the early stages, but not nearly as much, and white muscles reduce lipids.

Overall, the article is very well written and describes all the important aspects comprehensively.

However, I have some suggestions:

1. Please explain the reason of selecting 0, 6 and 24 hours for data collection, and why not 0, 6, 12 and 24 hours.

2. Please explain the importance of trypsin, amylase and lipase etc. in the introductory part

3. Line 192: Figure 1, Please properly abbreviate the “LT group”

4. Line 192: Figure 1, Why the results vary at 0 hour between the LT and HT groups? Please explain properly at what time exactly you recorded the data in minutes for 0 hour.

5. Explain why the amylase activity was lower as compared to other enzymes after 24 hours.

6. Figure 2: why there is huge difference among the enzymatic activity of different enzymes in liver. Please explain it properly in the results section. Why lactase dehydrogenase activity is higher than the rest?

7. Please correct the title of Figure 3 (L) as White muscle CEA. It would be better if you arrange the graphs A-D for red muscles, E-H as White muscles.

Reviewer #3: The paper is well written and adds to existing literature on the subject. The introduction is well composed and has been developed on the right lines. Most appropriate methodology has been used for analysis of data and information. The results have been reported well. Logical sequence of interpretation has been followed and developed scientifically. The discussion has been well brought out and the cogency of arguments are well thought out. The discussion is comprehensive and complete. References are as required. May be accepted for publication

6. PLOS authors have the option to publish the peer review history of their article (what does this mean?). If published, this will include your full peer review and any attached files.

Reviewer #1: **Yes: **Professor Daniel Ama-Abasi

Reviewer #2: **Yes: **Dr. Muhammad Adnan

Reviewer #3: No

While revising your submission, please upload your figure files to the Preflight Analysis and Conversion Engine (PACE) digital diagnostic tool, https://pacev2.apexcovantage.com/. PACE helps ensure that figures meet PLOS requirements. To use PACE, you must first register as a user. Registration is free. Then, login and navigate to the UPLOAD tab, where you will find detailed instructions on how to use the tool. If you encounter any issues or have any questions when using PACE, please email PLOS at figures@plos.org. Please note that Supporting Information files do not need this step.<quillbot-extension-portal></quillbot-extension-portal>

---

## [Author Response · Author response to Decision Letter 0]

9 Jul 2023

Reviewer #1: The experiment was well designed and the paper is very information. The significant and application of the study quite lucid. The following comments should however be noted for minor revision:

1. A sentence or two describing methodology should be inserted in the Abstract

Reply: The general method steps have been briefly added to the summary.” In this experiment, a heating rod was used to increase the temperature at 2℃/h to study the changes of energy allocation (CEA) and energy metabolity-related enzyme activities, including pepsin, trypsin, amylase, lipase, acid phosphatase, lactate dehydrogenase, alanine aminotransferase, glutamic oxalic aminotransferase and energy reserve (Ea), energy expenditure (ETS), in juvenile yellowfin tuna cells under acute temperature stress..” These are in lines 17-22 of the text.

2. Line 107- the intestines, pancreas, liver and blood were removed (not are removed from the fish).

Reply: Accept. The modified sentence reads: “The intestines, stomach, pyloric caeca, liver, red and white muscles and blood are removed.” These are in lines 120-121 of the text. 

3. Bulk of the discussion is a recapitulation of the results and methodology. Discussion should be majorly interpretation of results and inferences!

Reply: The description of the results has been simplified and the explanation of the results has been added. The specific part is discussed in the paper. The specific modifications are in lines 328-331, 409-412 and 415-422.

4. Why is cell division phase (prophase ) brought in here?

Reply: This word means the first 6 hours of the experiment, so it was deleted because it was not expressed properly. These are in lines 368 of the text.

5. Lates calcarifer being a species should be italicised, also Pelodiscus sinensis and Pelodiscus sinensi.

Reply: Species names have been italicized.

6. Chlamys Farreri should be written in italics-Chlamys ferreri, Gasterosteus aculeatus

Reply: Species names have been italicized.

7. In this study, intestinal amylase activity in the high temperature group decreased significantly over time and was lower than that in the control group, which was the same as the result of this study, trypsin increased significantly- recast this statement.

Reply: This sentence has been changed to “The changes of amylase in this study were similar to those in the Gasterosteus aculeatus. he activity of amylase is much lower than that of other enzymes because yellowfin tuna is a carnivorous fish and the activity of amylase is extremely low. It has been reported that the low energy activation of amylase activity in different fish is an adaptation of the digestive system to temperature maintenance by producing new highly efficient isoenzymes. Trypsin increased significantly, this is different from the above studies, which may be due to the unbalanced energy distribution of three-acanthus, which damages the synthesis of trypsin [52].” These are in lines 414-422 of the text.

Reviewer #2: This article entitled “Acute thermal stress increased enzyme activity and muscle energy distribution of yellowfin tuna” by Liu et al. is focused on impact of heat stress on enzyme activity of yellowfin tuna. Heat is a powerful stressor for fish living in natural and artificial environments. Understanding the effects of heat stress on the physiological processes of fish is essential for better aquaculture and fisheries management. The 24 h experiment was conducted to study the changes in cellular energy allocation (CEA) and energy metabolism-related enzyme activities of juvenile yellowfin tuna (Thunnus albacares) under acute temperature stress, which significantly reduced energy reserve (Ea) in muscle from 2 °C/h to high temperature (34 °C) from control (28 °C). It also increases energy expenditure (ETS). At 6 h, CEA decreased slightly in the high-temperature group, but, the difference in CEA between 24 h and 0 h decreased. After heat stress for 6 h, the activities of acid phosphatase (ACP), lactate dehydrogenase (LDH), alanine aminotransferase (ALT) and glutamic oxalacetic transaminase (AST) increased, indicating that the metabolic rate was accelerated. After heat stress for 24 h, the activity of ALT decreased, indicating that with time elapsed, the activities of some protein metabolizing enzymes increased, and some decreased. In this study, digestive enzymes, trypsin and lipase increased gradually. After heat stress, Ea and Ec change significantly. Yellowfin tuna muscles use lipids in response to sharp temperature increases at high temperatures, red muscles respond to temperature changes by increasing energy in the early stages, but not nearly as much, and white muscles reduce lipids.

Overall, the article is very well written and describes all the important aspects comprehensively.

However, I have some suggestions:

1. Please explain the reason of selecting 0, 6 and 24 hours for data collection, and why not 0, 6, 12 and 24 hours.

Reply：We pre-tested some indicators before doing the experiment, and found that the difference between 6h and 12h was not significant, which reduced the rate of fish loss. 

2. Please explain the importance of trypsin, amylase and lipase etc. in the introductory part

Reply: he importance of digestive enzymes has been added to lines 74-82 of the text.

3. Line 192: Figure 1, Please properly abbreviate the “LT group”

Reply: Changed to correct style. 

4. Line 192: Figure 1, Why the results vary at 0 hour between the LT and HT groups? Please explain properly at what time exactly you recorded the data in minutes for 0 hour.

Reply: Because the control group and the experimental group had their own fish. There are individual differences in physiological indexes among different individuals but no significant differences. The exact time for recording data is 6:00 PM. These are in lines 117-118 of the text.

5. Explain why the amylase activity was lower as compared to other enzymes after 24 hours.

Reply：Because yellowfin tuna is a carnivorous fish, amylase activity is extremely low. Lower-energy activation of amylase activity has been reported in different fish species as an adaptation of the digestive system to temperature maintenance through the production of new efficient isoenzymes. These are in 415-417 of the text.

6. Figure 2: why there is huge difference among the enzymatic activity of different enzymes in liver. Please explain it properly in the results section. Why lactase dehydrogenase activity is higher than the rest?

Reply: Because different enzymes have species and different effects. After the temperature rises, more lactic acid is secreted, so the lactate dehydrogenase is higher. The explanation has been inserted into lines 355-357 and 394-395 of the text.

7. Please correct the title of Figure 3 (L) as White muscle CEA. It would be better if you arrange the graphs A-D for red muscles, E-H as White muscles.

Reply: The ordinate name has been changed.

Reviewer #3: The paper is well written and adds to existing literature on the subject. The introduction is well composed and has been developed on the right lines. Most appropriate methodology has been used for analysis of data and information. The results have been reported well. Logical sequence of interpretation has been followed and developed scientifically. The discussion has been well brought out and the cogency of arguments are well thought out. The discussion is comprehensive and complete. References are as required. May be accepted for publication

Reply: Thank you for your recognition of our work！

---

## [Editor Report · Decision Letter 1]

24 Jul 2023

Acute thermal stress increased enzyme activity and muscle energy distribution of yellowfin tuna

PONE-D-23-13386R1

Dear Dr. Ma,

We’re pleased to inform you that your manuscript has been judged scientifically suitable for publication and will be formally accepted for publication once it meets all outstanding technical requirements.

Kind regards,

Mohammed Fouad El Basuini, Professor

Academic Editor

PLOS ONE

Additional Editor Comments (optional):

We’re delighted to let you know your manuscript has now been accepted for publication in PLOS ONE. The authors have made a reasonable and adequate revision to the manuscript. As such, we can accept your manuscript for publication. Congratulations

Reviewers' comments:

<quillbot-extension-portal></quillbot-extension-portal>

---

## [Editor Report · Acceptance letter]

27 Sep 2023

PONE-D-23-13386R1 

Acute thermal stress increased enzyme activity and muscle energy distribution of yellowfin tuna 

Dear Dr. Ma:

I'm pleased to inform you that your manuscript has been deemed suitable for publication in PLOS ONE. Congratulations! Your manuscript is now with our production department. 

Kind regards, 

on behalf of

Dr Mohammed Fouad El Basuini 

Academic Editor

PLOS ONE